# Differences in urine creatinine and osmolality between black and white Americans after accounting for age, moisture intake, urine volume, and socioeconomic status

**Patrick B. Wilson**●*, **Ian P. Winter, Josie Burdin**

Human Performance Laboratory, School of Kinesiology and Health Science, Old Dominion University, Norfolk, VA, United States of America

* pbwilson@odu.edu

**Data Availability Statement:** All relevant data are within the manuscript and its Supporting Information files.

## Abstract

Urine osmolality is used throughout research to determine hydration levels. Prior studies have found black individuals to have elevated urine creatinine and osmolality, but it remains unclear which factors explain these findings. This cross-sectional, observational study sought to understand the relationship of self-reported race to urine creatinine and urine osmolality after accounting for age, socioeconomic status, and fluid intake. Data from 1,386 participants of the 2009–2012 National Health and Nutrition Examination Survey were utilized. Age, poverty-to-income ratio (PIR), urine flow rate (UFR), fluid intake, estimated lean body mass (LBM), urine creatinine, and urine osmolality were measured. In a sex-specific manner, black and white participants were matched on age, dietary moisture, UFR, and PIR. Urine creatinine was greater in black men (171 mg/dL) than white men (150 mg/dL) and greater in black women (147 mg/dL) than white women (108 mg/dL) (p < .001). Similarly, urine osmolality was greater in black women than white women (723 vs. 656 mOsm/kg, p = .001), but no difference was observed between white and black men (737 vs. 731 mOsm/kg, p = .417). Estimated LBM was greater in black men (61.8 kg) and women (45.5 kg) than in white men (58.9 kg) and women (42.2 kg) (p≤.001). The strongest correlate of urine osmolality in all race-sex groups was urine creatinine (Spearman ρ = .68-.75). These results affirm that individuals identifying as black produce higher urine creatinine concentrations and, in women, higher urine osmolality after matching for age, fluid intake, and socioeconomic status. The findings suggest caution when comparing urine hydration markers between racial groups.

## Introduction

Avoiding hypohydration is a significant element of maintaining good overall health and physical function. Studies, for example, have documented that hypohydration can impair cognitive function [1] and increase perception of effort during exercise [2]. Also, while still not yet

**Funding:** The author(s) received no specific funding for this work.

**Competing interests:** The authors have declared that no competing interests exist.

confirmed as fully causal in nature, there is an observed relationship between levels of fluid-regulating hormones (i.e., arginine vasopressin) and metabolic function [3]. Given this evidence, the assessment of hydration status is broadly considered as important to researchers, clinicians, and practitioners.

A multitude of options exists for assessing hydration status, ranging from very practical/simple (urine color) to invasive (plasma osmolality). Experienced researchers in the field of hydration science routinely acknowledge that there is no single method that works best for every situation and that all assessment methodologies have advantages and disadvantages [4, 5]. However, given its relative ease of access and low cost, assessment of urine is frequently undertaken in both clinical and research settings. Of the three most common urine-based assessments available (color, specific gravity, and osmolality), urine osmolality and urine specific gravity (USG) are often viewed as the more valid options because evaluating urine color involves more sources of error (e.g., room lighting, evaluator experience, urine collection method) [6]. While there is variation in the literature, a threshold for urine osmolality that is frequently used to define hypohydration is ≥800 mOsm/kg [7, 8].

Based on a urine osmolality threshold of ≥800 mOsm/kg, it has been estimated that one-third of Americans are supposedly hypohydrated at any given time [9]. Furthermore, people identifying as black have been observed to have elevated urine osmolality, suggesting that the prevalence of hypohydration could be higher in this group [9, 10]. Studies showing lower water intakes among black individuals in the United States also support this hypothesis [10].

Recent research by Robinson et al. [11] suggests that these racial/ethnic differences in supposed hypohydration status are at least partly driven by differences in socioeconomic deprivation, which may ultimately impact the type and amount of fluid people consume. For example, black individuals may view tap water as less safe than whites do [12], a justifiable concern given the major water contamination crises that have occurred in black-majority localities like Flint, Michigan and Newark, New Jersey. However, another factor that may be playing a role in the higher rate of supposed hypohydration in blacks is fluid-independent variations in urine creatinine concentrations. Multiple large epidemiological studies have found that serum and urine creatinine are elevated in black individuals (e.g., [13, 14]), and while variances in fluid intake could provide an explanation for these findings, there is a strong basis for thinking that these differences may be partly independent of fluid intake. Specifically, serum creatinine increases in a linear fashion with African ancestry levels [15, 16]. In one study of UK Biobank participants, African ancestry levels explained over 70% of the variability in serum creatinine in men and women, and adjustment for socioeconomic deprivation did not attenuate the association [16].

With this literature in mind, this investigation's goal was to examine if racial differences in urine creatinine and a hydration biomarker (urine osmolality) were apparent in a sample of American adults after accounting for the influences of age, fluid intake, urine flow rate, and socioeconomic status. We hypothesized that individuals identifying as black would have higher urine creatinine concentrations and urine osmolality than whites, even after accounting for age, socioeconomic status, fluid intake, and urine flow rate.

## Methods

### Design and participants

The present study involved a secondary analysis of publicly available cross-sectional data from the National Health and Nutrition Examination Survey (NHANES). Survey years 2009–2012 were utilized, as urine osmolality was available only for that timespan. Files containing de-identified individual-level data were downloaded from the NHANES website. The NHANES

research protocols were reviewed and approved by the National Center for Health Statistics Ethics Review Board, and participants gave their written informed consent before participating. This study's protocol was submitted to the Human Subjects Review Committee of the College of Health Sciences at Old Dominion University and was determined to have exempt status.

For the 2009–2012 survey cycles, 26,215 individuals were screened, and 20,015 completed interviews and examinations (76% response rate). Additional exclusions for this analysis were made for individuals who were pregnant, those less than 20 years of age, and those who had missing values for variables of interest, including urine osmolality, urine creatinine, urine flow rate, anthropometrics, dietary data, and income-to-poverty ratio (PIR). In addition, this analysis relied on a smaller subset of black and white adult participants who were matched on important characteristics (age, fluid intake, urine flow rate, socioeconomic status). The matching process is described in detail later. Ultimately, 305 black men, 305 white men, 388 black women, and 388 white women were included in the analysis.

## Urine assessments

Spot urine samples were collected at mobile examination centers (MEC), with participants being asked to record the time of their last urination before arrival. Participants' MEC visits were randomly assigned to occur in the morning, afternoon, or evening. Participants were sent a reminder letter before their scheduled visit with instructions to record the time of their last urine void on a card. At the MEC visit, participants provided a urine sample, with instructions to completely empty their bladder into a container. Urine volume and collection time were recorded by study staff. Based on this information, flow rate of urine was calculated as follows: volume of urine / time (min) since last urination. This was then recalculated to a daily value (mL/24 h).

Urine osmolality was quantified with an OSMETTE II Model 5005, Automatic Osmometer (Precision Systems, Inc), which uses a freezing point depression method.

Urinary creatinine concentration was assessed utilizing a Roche/Hitachi Modular P Chemistry Analyzer, which employs an enzymatic (creatinase) method.

## Self-reported race

Background demographic information was collected in participants' homes during interviews. Self-reported race/ethnicity was based on series of questions about the participant's ethnic origins and racial identity. Responses to these questions were then used by NHANES personnel to group participants into one of the following five categories: Mexican American, other Hispanic, non-Hispanic white, non-Hispanic black, and other race or multi-racial. For the present analysis, only non-Hispanic black and white participants were included because differences in creatinine in previous studies have been most consistently observed in these two groups.

## Other variables

Regarding sex, NHANES uses the term *gender* in their documentation, but this analysis uses the descriptor *sex* because there are only two possible responses for the variable in the NHANES data files (male or female). Socioeconomic status was quantified via the PIR, which is the total income of the household where the participant lives divided by the income associated with poverty guidelines relative to family size, as well as the appropriate year and state. Values of greater than 5.0 were recoded as 5.0 by NHANES staff because of potential disclosure concerns.

Because of their potential to impact urine osmolality [17, 18], intakes of fluid (moisture in g/day), sodium (mg/day), and protein (g/day) were assessed with a 24-hour recall method that

is based on the United States Department of Agriculture Automated Multiple Pass Method. Dietary moisture reflects all water intake from foods and beverages. Of these three dietary variables, dietary moisture is likely the most important determinant of urine concentration and was selected as a variable for matching. The validity of the moisture intake estimates from this methodology is unclear [19]. Thus, we decided to not only match black and white participants on dietary moisture intake but also urine flow rate, because urine volume is a more objective measure and strongly associates with total fluid intake [20].

Anthropometric variables (body mass, height, waist circumference) were measured at in-person MEC visits. Because lean body mass (LBM) has been previously shown to associate positively with urine creatinine and markers of urine concentration [21, 22], validated equations were used to estimate LBM in kg [23]. The equations are as follows:

- *Men*: 0.001 (age in y) + 0.064 (height in cm) + 0.756 (mass in kg)– 0.366 (waist circumference in cm) + 19.363 + (0 for white; 0.432 for black)

- *Women*: -0.039 (age in y) + 0.186 (height in cm) + 0.383 (mass in kg)– 0.043 (waist circumference in cm)– 10.683 + (0 for white; 1.085 for black)

### Case-control matching

Matching of black and white participants was carried out using the case-control matching function in SPSS (version 29, IBM Corp, Armonk, NY). Variables matched on were age, dietary moisture, urine flow rate, and PIR, with tolerance values of 3 years, 200 g, 200 mL/24 h, and 0.5, respectively. These tolerance values were selected in order to ensure sample sizes of at least 300 for each group. Matching was carried out for men and women separately.

### Statistical analysis

Although NHANES data can be analyzed so that the estimates are nationally representative, the present analysis did not do that because of the case-control matching process, which resulted in a smaller subset of the original participants in the 2009–2012 NHANES. The distribution of variables was evaluated by inspecting histograms and Q-Q plots. Most of the variables, with the main exception of urine osmolality, showed a right-skewed distribution; consequently, descriptive statistics are reported using median ($25^{th}$-$75^{th}$ percentiles). Potential differences between black and white participants on matched variables (age, PIR, moisture intake, urine flow rate) and non-matched variables (urine osmolality, urine creatinine, estimated LBM, dietary protein, dietary sodium) were evaluated using Mann-Whitney U tests. The Spearman's rank-order correlation ($\rho$) was used to evaluate the strength of association between variables and urine osmolality. These correlations were carried out separately for black and white participants. Co-efficient $\rho$ sizes of 0.0–0.19, 0.20–0.39, 0.40–0.59, 0.60–0.79, and 0.8–1.0 were used to determine correlation degrees of very weak, weak, moderate, strong, and very strong, respectively. All analyses were done separated by sex. A two-sided $p < 0.05$ was used as the threshold for statistical significance.

### Results

Case-control matching resulted in samples of 305 black and 305 white men, as well as 388 black and 388 white women. Descriptive statistics for black and white participants by sex are reported in **Table 1**. As expected, there were no significant group differences for matched variables (age, PIR, moisture intake, urine flow rate), with p values all ≥0.87 for men and ≥0.89 for women. Further, dietary sodium and protein intakes did not differ significantly between

**Table 1. Descriptive statistics.**

| | Men | | | | Women | | |
|---|---|---|---|---|---|---|---|
| | **White (n = 305)** | **Black (n = 305)** | ***p*** | | **White (n = 388)** | **Black (n = 388)** | ***p*** |
| Age (years) | 53.0 (34.0–64.5) | 53.0 (33.5–63.5) | .912 | | 50.0 (36.0–63.0) | 50.0 (36.0–62.8) | .903 |
| PIR | 2.1 (1.2–4.5) | 1.9 (1.3–4.5) | .892 | | 1.6 (1.0–3.6) | 1.7 (1.0–3.8) | .999 |
| UFR (mL/24 h) | 893 (616–1,320) | 890 (625–1,300) | .873 | | 716 (477–1,037) | 719 (473–1,032) | .891 |
| Moisture (g) | 2,650 (2,136–3,227) | 2,671 (2,117–3,222) | .915 | | 2,174 (1,674–2,776) | 2,150 (1,647–2,754) | .916 |
| LBM (kg) | 58.9 (53.2–65.9) | 61.8 (53.8–70.5) | .001 | | 42.2 (37.2–47.5) | 45.5 (41.2–51.0) | < .001 |
| Dietary sodium (mg) | 3,728 (2,716–4,867) | 3,859 (2,768–5,082) | .616 | | 2,591 (1,897–3,554) | 2,790 (1,967–3,772) | .079 |
| Dietary protein (g) | 84 (64–111) | 90 (64–122) | .149 | | 60 (46–80) | 64 (45–86) | .184 |
| Urine creatinine (mg/dL) | 150 (99–200) | 171 (127–228) | < .001 | | 108 (68–160) | 147 (94–211) | < .001 |
| Urine osmolality (mOsm/kg) | 737 (559–877) | 731 (582–888) | .417 | | 656 (436–817) | 723 (516–878) | .001 |

LBM, lean body mass; PIR, poverty-to-income ratio; UFR, urine flow rate. Values are shown as median (25th-75th percentile).

black and white participants (all p values ≥.079). In contrast, the Mann-Whitney U tests identified statistically significant differences between black and white participants for estimated LBM (p≤.001) and urine creatinine (p < .001) in both men and women. Regarding urine osmolality, black women had higher values than white women (p = .001) but there was no racial difference in men (p = .417).

Among black men, there were significant correlations between urine osmolality and age ($\rho$ = -.39, p < .001), urine creatinine ($\rho$ = .68, p < .001), LBM ($\rho$ = .29, p < .001), and urine flow rate ($\rho$ = -.25, p < .001). The effect sizes of these relationships are weak for age, LBM, and urine flow rate, while the effect size is strong for urine creatinine. Among white men, there were significant correlations between urine osmolality and age ($\rho$ = -.23, p < .001), urine creatinine ($\rho$ = .70, p < .001), LBM ($\rho$ = .20, p < .001), and urine flow rate ($\rho$ = -.32, p < .001). As with black men, the effect sizes of these relationships for white men are weak for age, LBM, and urine flow rate, while the effect size is strong for urine creatinine. Moisture intake was not significantly associated with urine osmolality in men. Since urine creatinine was most strongly associated with urine osmolality, **Fig 1** shows the association between these two variables for black and white men separately, along with median values presented using horizontal and vertical lines.

Among black women, there were significant correlations between urine osmolality and age ($\rho$ = -.39, p < .001), urine creatinine ($\rho$ = .75, p < .001), LBM ($\rho$ = .17, p < .001), moisture intake ($\rho$ = -.11, p = .037), and urine flow rate ($\rho$ = -.31, p < .001). Effect sizes of these relationships are very weak for LBM and moisture intake, weak for age and urine flow rate, and strong for urine creatinine. Among white women, there were significant correlations between urine osmolality and age ($\rho$ = -.31, p < .001), urine creatinine ($\rho$ = .72, p < .001), LBM ($\rho$ = .17, p < .001), moisture intake ($\rho$ = -.17, p < .001), and urine flow rate ($\rho$ = -.27, p < .001). Effect sizes of these relationship for white women are very weak for LBM and moisture intake, weak for age and urine flow rate, and strong for urine creatinine. Since urine creatinine was most strongly associated with urine osmolality in women, **Fig 2** shows the association between these two variables for blacks and whites separately, along with median values presented using horizontal and vertical lines.

## Discussion

The main hypothesis of this study was that black individuals, as compared to their white counterparts, would produce higher urine creatinine concentrations and urine osmolality after

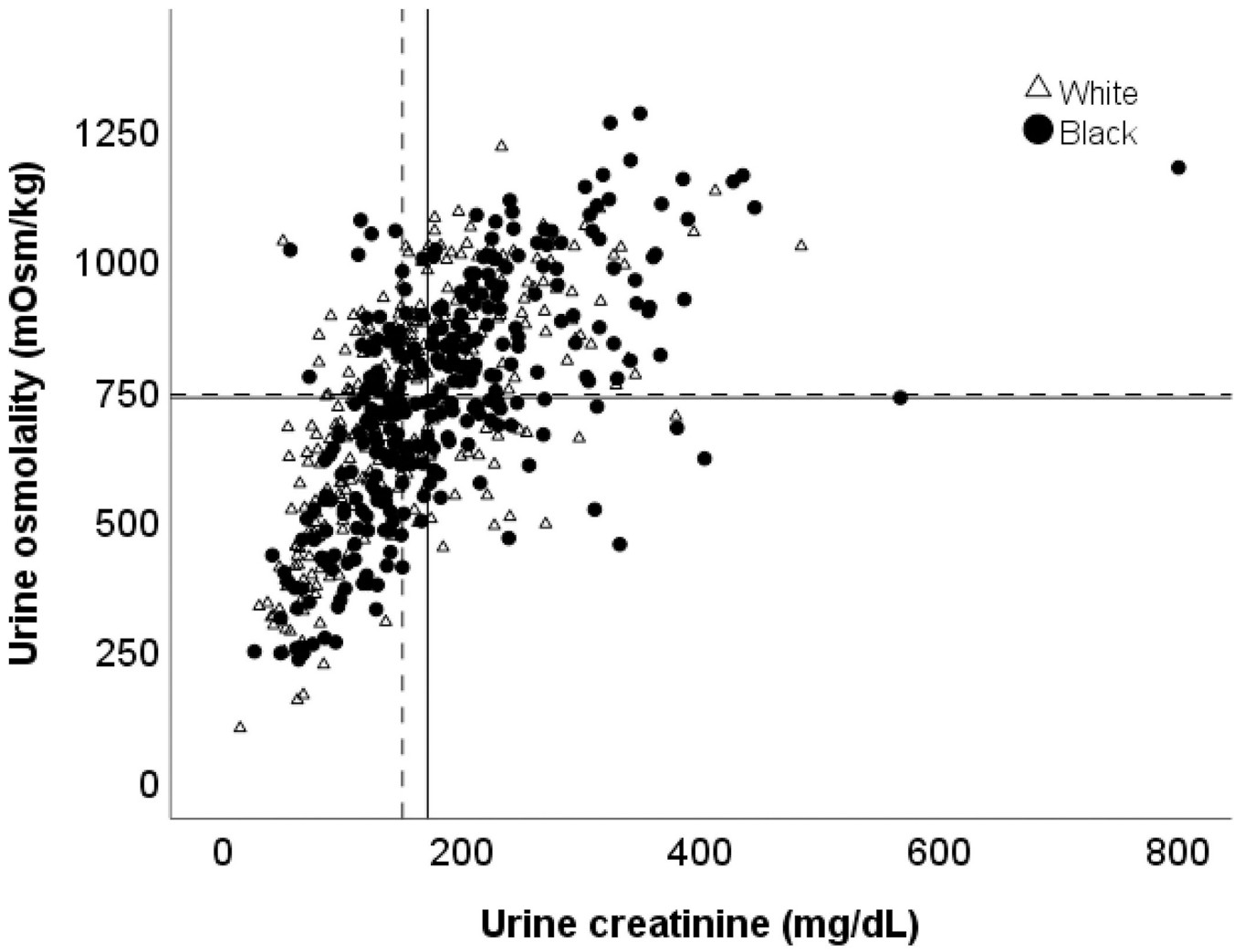

**Fig 1. Relationship between urine creatinine and urine osmolality in white (triangles) and black (circles) men.** Median values are represented by solid (black men) and dashed (white men) lines.

accounting for age, socioeconomic status, fluid intake, and urine flow rate. Indeed, black women were found to have significantly higher values of both urine osmolality and urine creatinine than white women. Black men were found to have significantly higher values of urine creatinine than white men, but there were no differences found in urine osmolality between black men and white men. The consistent finding of elevated urine creatinine in black women and men is most likely explained by the fact that increasing African ancestry levels are strongly and positively associated ($R^2 = 0.7$) with serum creatinine levels [16]. While self-reported race is not equivalent to measuring genetic ancestry, it is generally effective for classifying people into ancestral clusters [24]. In one study of women from New York City, for example, genetic ancestry levels were 77.6% African and 75.1% European for those who self-identified as black and white, respectively [25].

Black women had significantly higher urine osmolality compared to white women, while on the other hand, there was no significant difference in urine osmolality between black men and white men. This could be due to the relative sex differences in urine creatinine between black and white participants. Specifically, the median urine creatinine for black women was

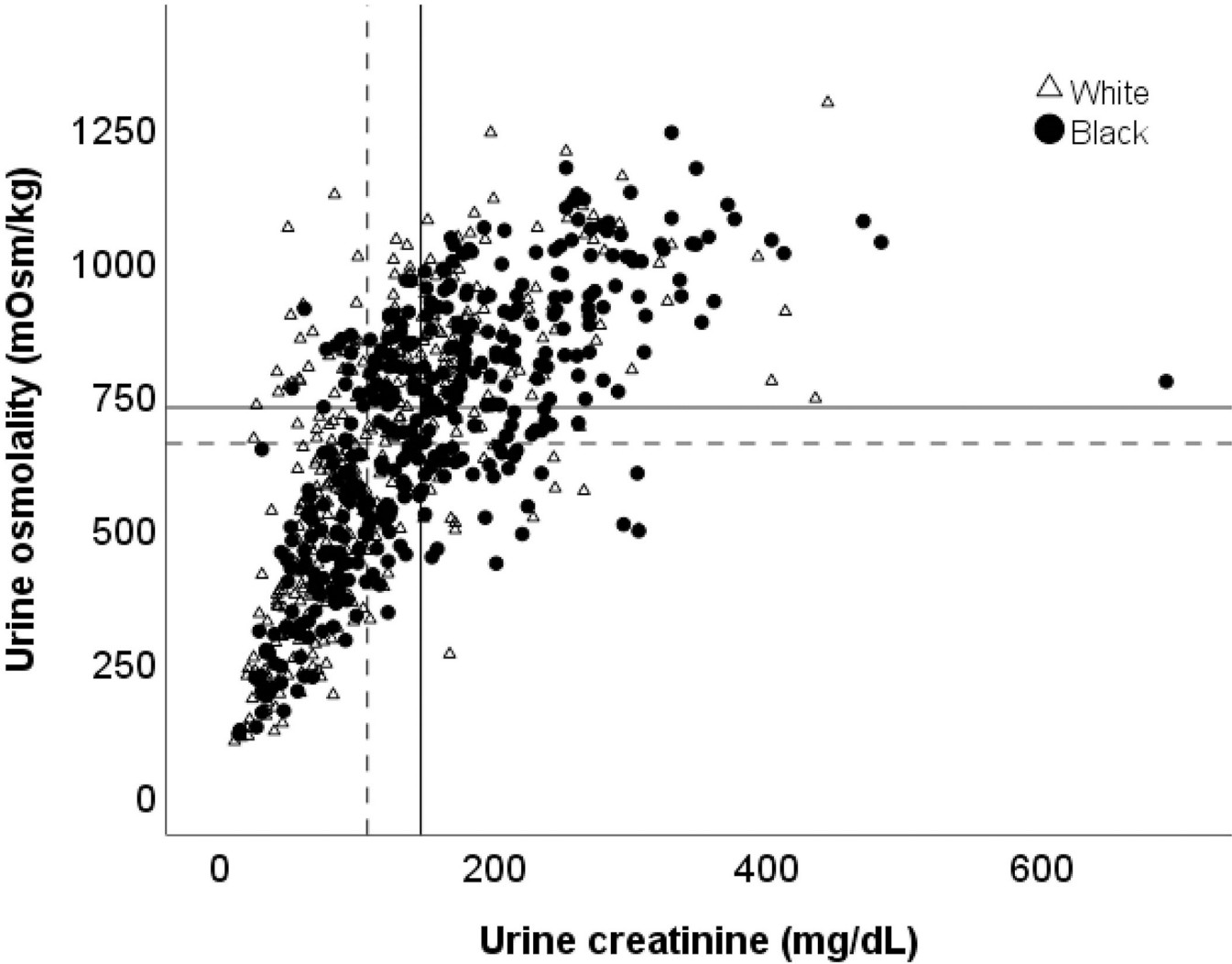

**Fig 2. Relationship between urine creatinine and urine osmolality in white (triangles) and black (circles) women.** Median values are represented by solid (black women) and dashed (white women) lines.

36% higher than for white women, while this relative racial difference was only 14% among men. A potential explanation for the larger relative urine creatinine difference in women is that median estimated LBM was 7.8% higher in black women relative to white women, while for black and white men this relative difference in LBM was apparently smaller (4.9%). Creatinine is a byproduct of muscle metabolism and generally associates strongly with total amounts of skeletal muscle mass [26], a major component of LBM. Thus, a larger relative difference in LBM between black and white women could translate to a larger relative difference in urine creatinine as compared to men.

Although urine osmolality did not vary significantly between black and white men (even with a difference in urine creatinine), it is possible that other urine-based measures of hydration status may differ by self-reported race. USG, for instance, is a common measure of hydration status that tends to be more frequently used in field settings than urine osmolality due to the availability of portable, relatively inexpensive urine refractometers. Notably, as compared to urine osmolality, USG is more substantially impacted by variations in urine creatinine [27].

This is because USG is influenced by both molecule number and size, while urine osmolality depends only on molecule number [27]. The relatively large molecular weight of creatinine means that any increases in creatinine will raise USG more than urine osmolality. In support of this contention, a prior study of NHANES showed that black individuals were more likely than whites to have an elevated USG (>1.02) [28]. Unfortunately, USG is not available for the 2009–2012 years of NHANES, meaning we were unable to address this possibility directly.

Despite significant correlations between several variables of interest, only urine creatinine had a large effect size with urine osmolality. Other factors with a weaker relationship to urine osmolality included age, LBM, and urine flow rate for both sexes and races, and moisture intake for females of both races. These findings emphasize the strong relationship between urine creatinine and urine osmolality. Additionally, it further supports the idea that any factor that is associated with higher creatinine—including self-reported black race or higher African ancestry—is a potentially important determinant of urine osmolality.

Despite having a weaker effect size, estimated LBM was statistically significant in its correlation with urine osmolality. The relatively weak association ($\rho$ = 0.17–0.29) is likely due to using an estimated value of LBM derived from predictive equations [23]. It is possible that LBM and osmolality would have had a stronger association if more direct measures of LBM were taken. Additionally, muscle mass is just one of the components of LBM, with additional components such as water weight, organ tissue, and bone mass. This could have also contributed to the weaker association between LBM and osmolality as compared to what would be expected between muscle mass and osmolality.

Another finding of this study is that black individuals had higher amounts of LBM than whites, which could at least partially explain the racial differences in urine creatinine. Indeed, some limited research supports the notion that having a high African ancestry admixture is associated with lower fat mass and higher LBM [29, 30]. Thus, the previously noted positive association between African ancestry and concentrations of serum creatinine [16] may be partly driven by racial differences in LBM.

Another possibility to consider is that the higher LBM observed among black participants may reflect a truly greater daily water requirement. In adults, fat-free mass is typically comprised of 70–75% water [31], meaning that those with greater amounts of LBM could theoretically require greater water intakes to maintain their body water stores. Since we did not match on LBM, it is therefore possible that the higher LBM and urine creatinine observed in black participants reflects a truly greater water need. However, other research has shown that the variance in daily water turnover is not explained well by anthropometric variables like weight, height, or body mass index [32], and the necessity of taking body size into consideration when making fluid intake recommendations for adults is uncertain [31].

The results of this research have allowed further insight into the differences in urine creatinine and osmolality between black and white Americans. However, there are a few limitations of the research to take into consideration. While urine osmolality and USG are both commonly used to assess hydration status, this specific study only considered urine osmolality due to a lack of USG data for years 2009–2012. Implementing both USG and urine osmolality assessments while evaluating urine creatinine would allow for a more detailed assessment of how urine creatinine impacts both of these urine markers in different racial groups. Another limitation of the study is the assessment of urine flow rate. Participants were reminded to record the time of their last urine void before their scheduled visit. This time was used in the urine flow rate calculation, which is easily subject to error. For example, if participants estimated the time this occurred instead of recording it in real time, the urine flow rate calculation may have been altered significantly, negatively affecting the present results. In addition, the use of spot urine sampling, which is the method employed by NHANES, is not as valid as

24-hour urine collections, especially when it comes to being used as an indicator of daily fluid intake [20].

## Conclusion

Prior research has found elevated urine osmolality in individuals self-identifying as black [9, 10], but the reasons for this difference remain under-investigated. Although studies support the idea that fluid intakes are lower among individuals self-identifying as black in the United States, and that socioeconomic deprivation likely plays an important role in this disparity, previous research has also found that serum creatinine levels increase in those with higher levels of African ancestry, even when adjusted for socioeconomic deprivation [16]. The present analysis extends these prior results to a large sample of American adults, including black and white women and men matched on age, fluid intake measures (dietary moisture and urine flow rate), and socioeconomic status. Overall, urine creatinine was elevated among black men and women compared to whites after accounting for the previously listed influencing factors, although the effect was larger in women and a difference in urine osmolality was not observed in men. Further research should be conducted to examine whether the greater LBM and urine creatinine observed in black participants reflect an actual increase in water requirements. For the time being, our results suggest that researchers and practitioners should use caution when directly comparing urine hydration markers between racial groups.

## Supporting information

**S1 File. Dataset.**
(XLS)

## Author Contributions

**Conceptualization:** Patrick B. Wilson.

**Formal analysis:** Patrick B. Wilson, Ian P. Winter.

**Methodology:** Patrick B. Wilson.

**Supervision:** Patrick B. Wilson.

**Visualization:** Patrick B. Wilson, Ian P. Winter.

**Writing – original draft:** Patrick B. Wilson, Ian P. Winter, Josie Burdin.

**Writing – review & editing:** Patrick B. Wilson, Ian P. Winter, Josie Burdin.

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
