## [Decision Letter · Decision Letter 0]

23 Apr 2024

PONE-D-24-10364Differences in urine creatinine and osmolality between black and white Americans after accounting for age, moisture intake, urine volume, and socioeconomic statusPLOS ONE

Dear Dr. wilson,

Thank you for submitting your manuscript to PLOS ONE. After careful consideration, we feel that it has merit but does not fully meet PLOS ONE’s publication criteria as it currently stands. Therefore, we invite you to submit a revised version of the manuscript that addresses the points raised during the review process.

We look forward to receiving your revised manuscript.

Kind regards,

William M. Adams

Academic Editor

PLOS ONE

Journal Requirements:

3. We are unable to open your Supporting Information file "Final dataset_matched_blackwhite_bothsexes.sav". Please kindly revise as necessary and re-upload.

Reviewers' comments:

Reviewer's Responses to Questions

**Comments to the Author**

1. Is the manuscript technically sound, and do the data support the conclusions?

Reviewer #1: Yes

Reviewer #2: Partly

2. Has the statistical analysis been performed appropriately and rigorously? 

Reviewer #1: Yes

Reviewer #2: Yes

3. Have the authors made all data underlying the findings in their manuscript fully available?

Reviewer #1: Yes

Reviewer #2: Yes

4. Is the manuscript presented in an intelligible fashion and written in standard English?

Reviewer #1: Yes

Reviewer #2: Yes

5. Review Comments to the Author

Reviewer #1: I read with great interest this manuscript and I would like to congratulate with the authors for this well-done work. The manuscript is well-written, the topic is of interest and the discussion provides severaly hypotheses explaining the reported findings.

I have only a question: did the authors consider salt intake as a variable that might affect hydration parameters and possible differences among diet between the two selected populations? Despite matching for social-economic status could have reduced differences between black and white people, do you think that black individuals might be characterized by a higher salt intake (Yoon et al., 2024)? Maybe another point of discussion might be related to renal handling and reabsorption of salt and water, if differences are present based on racial/ethnic differences.

Reviewer #2: In this manuscript, the authors use NHANES data combined from 2009-2012 to examine differences in hydration status (urine osmolality) and urine creatinine. Based on higher urine osmolality in black women compared to white women and correlations between urine osmolality and urine creatinine among racial groups, the authors recommend caution when comparing urine hydration markers between racial groups.

Major Comments:

I am not sure that the case matching makes sense in this case. Can the authors better describe why each group was matched for both moisture intake and UFR simultaneously? I would expect these variables to be correlated with one another, in which case wouldn’t it make sense to only match by UFR if the authors believe that is more representative of the actual fluid intake of participants? https://www.ncbi.nlm.nih.gov/pmc/articles/PMC2827892/

Can the authors also further describe how the case-control matching was performed (i.e., how was this specific subset generated)? Particularly because the sample size has reduced considerably to achieve this matching.

Dietary salt intake may also influence creatinine clearance and appears to be something that should be accounted for: https://pubmed.ncbi.nlm.nih.gov/35157527/

Is LBM not also associated with urinary creatinine? It may be that the LBM differences (and perhaps physical activity as well) are driving these differences in creatinine. You acknowledge this in your female group as this possibly accounting for the differences. The study you cite regarding the genetic ancestry differences contributing to creatinine differences included BMI but not lean body mass in the model. So could it actually be the LBM driving these differences?

The lean body mass differences between each group would likely necessitate greater fluid intake between groups. Thus, because fluid intake was equated, it seems natural that there is higher osmolality in the group with higher fluid intake requirements, at least in the case of the females where this difference was more apparent.

I am not sure how the conclusion has been reached that caution should be used when directly comparing urine hydration markers and rates of hypohydration between racial groups. Based on the results presented, it would seem more accurate to conclude when accounting for age, PIR, UFR, and moisture intake that there are no significant differences in urine osmolality between men of different races. However, when accounting for these factors in females, urine osmolality remained higher in black females, perhaps related to the greater magnitude of LBM and perhaps fluid requirements of those specific females. Also, the “rates” of hypohydration were not examined in this study (for example, by using the cutoff of >800mOsm/kg), so I think it may be a stretch to include this as part of the conclusion.

Minor comments:

Lines 60-68 – I do not think urine specific gravity needs to be described here since it is not mentioned elsewhere in the results. At least not in terms of its commonly used cutoff point.

Can the authors also briefly comment somewhere in the limitations on the drawbacks of spot urine samples vs 24hr urine samples?

6. PLOS authors have the option to publish the peer review history of their article (what does this mean?). If published, this will include your full peer review and any attached files.

Reviewer #1: No

Reviewer #2: No

---

## [Author Response · Author response to Decision Letter 0]

8 May 2024

We appreciate the reviewers’ time and effort providing feedback on our manuscript. We have used their helpful comments and suggestions to improve the overall quality of the manuscript. Each of the reviewer’s comments is listed below, followed by our responses.

Reviewer #1

I read with great interest this manuscript and I would like to congratulate with the authors for this well-done work. The manuscript is well-written, the topic is of interest and the discussion provides severaly hypotheses explaining the reported findings.

• Thank you for taking the time to review our manuscript. We appreciate the positive comments regarding our work. 

I have only a question: did the authors consider salt intake as a variable that might affect hydration parameters and possible differences among diet between the two selected populations? Despite matching for social-economic status could have reduced differences between black and white people, do you think that black individuals might be characterized by a higher salt intake (Yoon et al., 2024)? Maybe another point of discussion might be related to renal handling and reabsorption of salt and water, if differences are present based on racial/ethnic differences.

• This is an interesting question. We did consider including sodium in the analysis during the initial planning stages of the project. However, based on some previous work from our lab that showed sodium was not a major determinant of urine specific gravity in American adults (https://pubmed.ncbi.nlm.nih.gov/34470907/), we ultimately decided to omit it from the original analysis. That being said, we agree with the reviewer that it may be valuable for readers to see the data on dietary sodium. Thus, we have added the median (25-75th percentile) values along with Mann Whitney U test results to Table 1. In both men and women, there were no statistically significant differences in sodium intakes between black and white individuals. We also added the statistics for dietary protein, since it could potentially impact urine osmolality via the production of urea. Similar to sodium, there were no statistically significant differences in protein intakes between black and white individuals.

Reviewer #2: 

In this manuscript, the authors use NHANES data combined from 2009-2012 to examine differences in hydration status (urine osmolality) and urine creatinine. Based on higher urine osmolality in black women compared to white women and correlations between urine osmolality and urine creatinine among racial groups, the authors recommend caution when comparing urine hydration markers between racial groups.

Major Comments:

I am not sure that the case matching makes sense in this case. Can the authors better describe why each group was matched for both moisture intake and UFR simultaneously? I would expect these variables to be correlated with one another, in which case wouldn’t it make sense to only match by UFR if the authors believe that is more representative of the actual fluid intake of participants? https://www.ncbi.nlm.nih.gov/pmc/articles/PMC2827892/

• Thank you for the insightful comment/question. In general, we decided to use the case-control matching design for a couple of reasons. The primary reason is that we find that, in comparison to some other statistical techniques, the results from case-control matching are fairly simple for most readers (including non-scientist practitioners) to understand and interpret. For example, multivariate regression may have some advantages over case-control matching, particularly from a statistical power perspective, but the output is more difficult for many practitioners to comprehend. In contrast, with case-control matching, the reader can easily understand that we made the two groups roughly equivalent on the matched variables. 

• As to why we decided to match on both moisture intake and urine volume, we feel that it does provide a modestly higher degree of rigor than matching on one or the other. Yes, we agree that urine volume is likely to correlate relatively well with total fluid intake, but the correlations can range from 0.3 to 0.8 depending on whether 24-h vs. spot urine collections are used (https://www.nature.com/articles/ejcn201393;
https://www.ncbi.nlm.nih.gov/pmc/articles/PMC8411265/). In addition, it is certainly possible that a person could have a high fluid intake but still have a low-to-moderate urine output due to high sweat losses. While we acknowledge that that scenario likely represents a minority of people, we still feel that matching on both dietary moisture intake and urine volume is the most robust approach. 

Can the authors also further describe how the case-control matching was performed (i.e., how was this specific subset generated)? Particularly because the sample size has reduced considerably to achieve this matching.

• Case-control matching was carried out with SPSS software. An example of how the process works can be viewed in the following video: https://www.youtube.com/watch?v=E6nzTTGIWss. In our study, we selected 4 variables to match white and black participants on (age, PIR, moisture intake, and urine flow rate). These four variables were selected because they are potential confounders of the relationship between race and our outcome urine measures (creatinine and osmolality). Tolerance/fuzz values are set for each variable, which means that each member of a matched pair needs to have values for a matched variable within the range of tolerance set. 

• As pointed out by the reviewer, we acknowledge that the case-control matching process reduced our sample size. However, all our sex-race subgroups still had a sample size of >300, which is large enough to detect relatively small effect sizes. A bigger sample size could potentially yield more statistically significant effects, but the practical relevance of those very small effects would be questionable. 

Dietary salt intake may also influence creatinine clearance and appears to be something that should be accounted for: https://pubmed.ncbi.nlm.nih.gov/35157527/

• Thanks for the suggestion. Reviewer 1 also requested that we take into consideration dietary sodium intake. We have added median values of sodium intake to Table 1. In both sexes, there were no statistically significant differences in sodium intake between black and white participants.

Is LBM not also associated with urinary creatinine? It may be that the LBM differences (and perhaps physical activity as well) are driving these differences in creatinine. You acknowledge this in your female group as this possibly accounting for the differences. The study you cite regarding the genetic ancestry differences contributing to creatinine differences included BMI but not lean body mass in the model. So could it actually be the LBM driving these differences?

• Yes, we do think that it is likely that some of the observed racial differences in urine creatinine are being driven by LBM differences, or more precisely, differences in skeletal muscle mass (which is not available in the NHANES dataset). We briefly mentioned this in our Discussion section (“Another finding of this study is that black individuals had higher amounts of LBM than whites, which could partly explain the racial differences in urine creatinine”), but we have added some additional text to the Discussion about this possibility. Specifically, some limited research supports the notion that having a high African ancestry admixture is associated with lower fat mass and higher lean mass. Thus, the association between African ancestry and higher concentrations of serum creatinine may be at least partly driven by racial differences in lean mass.

The lean body mass differences between each group would likely necessitate greater fluid intake between groups. Thus, because fluid intake was equated, it seems natural that there is higher osmolality in the group with higher fluid intake requirements, at least in the case of the females where this difference was more apparent.

• We see where the reviewer is coming from, but we do not necessarily agree that having a higher lean body mass necessitates a higher fluid intake. In multiple population studies, there is little-to-no relationship between BMI (or body size) and urine volume, meaning that larger individuals generally probably aren’t naturally choosing to consume larger amounts of fluid on a daily basis than smaller individuals (see example studies listed below). We realize that BMI is not the same as LBM, but the two measures tend to correlate moderately with one another, at least when quantified as absolute amounts (total kg). Furthermore, a study by Raman et al. (2004) reported that anthropometric variables explained little of the variance in daily water turnover in adults (https://journals.physiology.org/doi/full/10.1152/ajprenal.00295.2003). In total, there is not clear evidence that fluid intake requirements or water turnover consistently depend on body size. Although we agree that there could in theory be a higher daily fluid intake requirement for larger individuals with more LBM, that idea is still up for debate. Nonetheless, we have added the following text to the Discussion section to address the issue raised by the reviewer. “Another possibility to consider is that the higher LBM observed among black participants may reflect a truly greater daily water requirement. In adults, fat-free mass is typically comprised of 70-75% water [31], meaning that those with greater amounts of LBM could theoretically require greater water intakes to maintain their body water stores. Since we did not match on LBM, it is therefore possible that the higher LBM and urine creatinine observed in black participants reflects a truly greater water need. However, other research has shown that the variance in daily water turnover is not explained well by anthropometric variables like weight, height, or body mass index [32], and the necessity of taking body size into consideration when making fluid intake recommendations for adults is uncertain [31].”

• https://journals.lww.com/CJASN/fulltext/2011/11000/Urine_Volume_and_Change_in_Estimated_GFR_in_a.14.aspx

• https://link.springer.com/article/10.1186/1471-2369-14-246

• https://ehp.niehs.nih.gov/doi/full/10.1289/ehp.1408944

I am not sure how the conclusion has been reached that caution should be used when directly comparing urine hydration markers and rates of hypohydration between racial groups. Based on the results presented, it would seem more accurate to conclude when accounting for age, PIR, UFR, and moisture intake that there are no significant differences in urine osmolality between men of different races. However, when accounting for these factors in females, urine osmolality remained higher in black females, perhaps related to the greater magnitude of LBM and perhaps fluid requirements of those specific females. Also, the “rates” of hypohydration were not examined in this study (for example, by using the cutoff of >800mOsm/kg), so I think it may be a stretch to include this as part of the conclusion.

• We appreciate the reviewer’s perspectives on our conclusions. Taking their comments into consideration, we have removed reference to “rates of hypohydration.” In addition, we have added a sentence highlighting the need for additional research on whether the greater amounts of LBM and urine creatinine observed in black participants reflect an actual increase in water requirements.

Minor comments:

Lines 60-68 – I do not think urine specific gravity needs to be described here since it is not mentioned elsewhere in the results. At least not in terms of its commonly used cutoff point.

• We have edited this section as recommended. 

Can the authors also briefly comment somewhere in the limitations on the drawbacks of spot urine samples vs 24hr urine samples?

• We have added some text to the limitations section regarding the use of spot urine samples.

---

## [Decision Letter · Decision Letter 1]

20 May 2024

Differences in urine creatinine and osmolality between black and white Americans after accounting for age, moisture intake, urine volume, and socioeconomic status

PONE-D-24-10364R1

Dear Dr. wilson,

We’re pleased to inform you that your manuscript has been judged scientifically suitable for publication and will be formally accepted for publication once it meets all outstanding technical requirements.

Kind regards,

William M. Adams

Academic Editor

PLOS ONE

Additional Editor Comments (optional):

Reviewers' comments:

Reviewer's Responses to Questions

**Comments to the Author**

1. If the authors have adequately addressed your comments raised in a previous round of review and you feel that this manuscript is now acceptable for publication, you may indicate that here to bypass the “Comments to the Author” section, enter your conflict of interest statement in the “Confidential to Editor” section, and submit your "Accept" recommendation.

Reviewer #1: All comments have been addressed

Reviewer #2: All comments have been addressed

2. Is the manuscript technically sound, and do the data support the conclusions?

Reviewer #1: Yes

Reviewer #2: Yes

3. Has the statistical analysis been performed appropriately and rigorously? 

Reviewer #1: Yes

Reviewer #2: Yes

4. Have the authors made all data underlying the findings in their manuscript fully available?

Reviewer #1: Yes

Reviewer #2: Yes

5. Is the manuscript presented in an intelligible fashion and written in standard English?

Reviewer #1: Yes

Reviewer #2: Yes

6. Review Comments to the Author

Reviewer #1: I would like to thank the authors for addressing my comment and I am totally satisfied with their responses.

Thank you for adding this information on salt and protein intake.

Reviewer #2: All comments have been addressed well by the authors. This manuscript appears suitable for publication in its present form. Well done!

7. PLOS authors have the option to publish the peer review history of their article (what does this mean?). If published, this will include your full peer review and any attached files.

Reviewer #1: No

Reviewer #2: No

---

## [Editor Report · Acceptance letter]

22 May 2024

PONE-D-24-10364R1 

PLOS ONE

Dear Dr. wilson, 

I'm pleased to inform you that your manuscript has been deemed suitable for publication in PLOS ONE. Congratulations! Your manuscript is now being handed over to our production team.

Kind regards, 

on behalf of

Dr. William M. Adams 

Academic Editor

PLOS ONE